# Quality of Life and Depressive Symptoms in Transcatheter Aortic Valve Implementation Patients—A Cross-Sectional Study

**DOI:** 10.3390/healthcare10112211

**Published:** 2022-11-03

**Authors:** Katarzyna Olszewska-Turek, Barbara Bętkowska-Korpała

**Affiliations:** 1Medical Psychology Department, Chair of Psychiatry, Jagiellonian University Medical College, 31-501 Kraków, Poland; 2Department of Clinical Psychology, University Hospital, 30-688 Kraków, Poland; 3Department of Cardiology and Cardiovascular Interventions, University Hospital, 30-688 Kraków, Poland

**Keywords:** depression, health-related quality of life (HRQoL), transcatheter aortic valve implementation (TAVI), older adults, aortic valve stenosis

## Abstract

Quality of life is an important factor influencing mood. In any group of elderly people undergoing valve implementation or surgical aortic valve replacement, one in three will have depressive symptoms. The aims of this study were as follows: 1. to evaluate the impact of health-related quality of life on depressive symptoms in elderly patients undergoing TAVI, and 2. to analyze beliefs about TAVI. Methods: A total of 131 elderly people (mean age: 82.1 ± 6.1 years) scheduled to receive TAVI completed the Geriatric Depression Scale, EQ-5D-3L, and Mini Mental State Examination. A total of 43 patients completed the questionnaires after the treatment. The narrative interview analyses were performed based on 20 randomly selected patients after TAVI. Results: The mean level of general depression before TAVI was 4.19 ± 2.83, and after it was 3.12 ± 2.52 (*p* = 0.02), and the frequency decreased from 20% to 3%. An increase in the level of activity and number of interests and a decrease in life satisfaction were identified. The higher the general quality of life was, the lower the levels of depressiveness before and after TAVI were (r = 0.26 vs. r = 0.48; *p* < 0.05). Conclusions: Patients differed in their depressive symptoms, as well as their intensity and frequency, before and after TAVI. These results underscore the importance of screening for depression at baseline and reassessing changes in depressiveness during follow-up.

## 1. Introduction

The prevalence of major depressive disorders in the general population in Europe is estimated to be 6.6–6.9% [1,2]. It reaches 9.1% in older people (≥65) in the EU [1] and 15–30% in patients with coronary artery disease [3]. In a group of older adults undergoing transcatheter aortic valve implementation (TAVI) or surgical aortic valve replacement (SAVR), one in three has depressive symptoms [4,5]. These data are in line with the results of the PolSenior study, in which clinically significant depressive symptoms were found in 29.7% of the elderly population aged 65 and over in Poland [6]. The European Society of Cardiology and the American Heart Association list depression as a modifiable risk factor for coronary artery disease and for major adverse cardiovascular events after the onset of acute coronary syndromes, among other risk factors, such as hypertension, smoking, and hyperlipidemia [3,7,8,9,10,11]. Recent studies have shown that persistent depression is associated with a 3-fold increase in mortality at 6 months, while baseline depression is associated with mortality at 1 month and at 12 months among the elderly undergoing TAVI treatment [5]. Furthermore, cognitive impairment and depressive symptoms are two risk factors associated with an increased risk of mortality in a progressive, additive manner in elderly adults [10,11,12]. Previous research has shown that TAVI improved the health-related quality of life (HRQoL) in multiple dimensions, such as, mobility, self-care, and pain perception, among others [13,14,15,16,17,18], and reduced depression and anxiety levels in patients with aortic stenosis [19,20]. The population which benefits from TAVI is increasing. Consequently, elderly patients with aortic stenosis and many comorbidities, who previously were inoperable, are successfully admitted for this procedure [21]. Nevertheless, little is known about the role of depression in older adults undergoing transcatheter aortic valve implementation, especially with respect to results based on quality measures and interview methods such as narration and thematic analysis. Several previous studies have emphasized that the diagnosis should be confirmed through a comprehensive expert evaluation in addition to the recommendation of the American Heart Association to screen for depression using a brief questionnaire [3,5,12,22].

As clinicians should be aware that the diagnosis of depression should not be solely based on questionnaire instruments, the aim of this study is to present a quantitative (via the questionnaire method) and qualitative (via narrative interview and thematic analysis) analysis of the mood characteristics of older adults undergoing TAVI. The additional advantage of this study is its presentation of a qualitative analysis in the form of a narrative analysis of the beliefs of older people who have undergone TAVI. The hypothesis presented is that the severity of depressive symptoms is related to the level of health-related quality of life in TAVI patients.

## 2. Materials and Methods

### 2.1. Participants

A total of 131 patients with severe symptomatic aortic stenosis (AS) were examined during the qualification process for TAVI treatment. The patients were between 61 and 94 years old (mean age: 82.1 ± 6.1 years), and 63% (83) were female. The females and males did not differ in average age (82.5 ± 5.65 vs. 81.3 ± 6.76; *p* = 0.24) or mental state level (25.4 ± 3.98 vs. 25.73 ± 4.17; *p* = 0.63). The mean level of the general cognitive status according to the MMSE showed no mental impairment in the studied patients before and after TAVI (25.5 ± 4.05 points vs. 25.4 ± 3.46; *p* = 0.93). The level of education was significantly higher among the males than the females (11.13 ± 5.95 vs. 8.66 ± 3.35 years; *p* < 0.003). A group of 43 patients treated with TAVI filled the questionnaires and participated in a psychological narrative interview approximately one year after the treatment (Figure 1). The demographic and clinical characteristics are presented in Table 1. The qualification process, according to the inclusion and exclusion criteria, was presented in a previous paper [13]. The exclusion criteria included: a lack of informed consent, severe dementia, a severe somatic state preventing participation in the study, and an eyesight or hearing impairment precluding the completion of the questionnaires or answering of the interview questions.

### 2.2. Methods

The Geriatric Depression Scale-Short Form (GDS-SF), EQ-5D-3L questionnaire, and Mini-Mental State Examination Scale (MMSE) results were collected. In addition, the patients were asked to participate in a psychological narrative interview concerning their current life situation, with a particular interest in their everyday mood, and an interview concerning their demographic data, previous mental and somatic problems, and treatment. The study was cross-sectional and conducted in two stages, before and approximately thirteen months after TAVI. All the patients provided written informed consent to participate in the study. The protocol was approved by the local Bioethics committee of Jagiellonian University (decision no. 122.6120.39.2015). The study was conducted in accordance with the ethical principles for clinical research based on the Declaration of Helsinki with later amendment.

The following research tools were used:The Geriatric Depression Scale-Short Form (GDS-SF): a short form of the Geriatric Depression Scale, consisting of 15 items, which has been shown to be a reliable measure of depressive symptoms in older adults, with GDS-SF ≥ 6 cut-off level for depression. Cronbach’s alpha coefficient for GDS-SF is 0.92 [23,24]. GDS-SF consists of the following item topics: GDS-SF total—level of depressiveness; GDS-SF 1—satisfaction with life; GDS-SF 2—activities and interests; GDS-SF 3—emptiness of life; GDS-SF 4—boredom; GDS-SF 5—good spirits; GDS-SF 6—fear; GDS-SF 7—sense of happiness; GDS-SF 8—feeling of helplessness; GDS-SF 9 –preference for staying at home instead of going out; GDS-SF 10—memory problems; GDS-SF 11—excitement to be alive; GDS-SF 12—sense of worthlessness; GDS-SF 13—level of energy; GDS-SF 14—sense of hopelessness; GDS-SF 15—belief in being worse than others.EQ-5D-3L questionnaire, one of the most widely used methods for measuring health-related quality of life (HRQoL). It consists of two parts: the EQ-5D descriptive system and the EQ-5D visual analogue scale (EQ VAS). The EQ-5D-3L comprises five dimensions: mobility (MO), self-care (SC), usual activities (UA), pain/discomfort (PD), and anxiety/depression (AD). Within each of the dimensions, one of the three levels of functioning can be chosen (1 for “no problems”, 2 for “some problems”, and 3 for “extreme problems”). The EQ VAS part represents the responder’s self-rated health, marked on a vertical thermometer-like visual analogue scale, where the end point labelled “0” means “The worst health you can imagine” and “100” means “The best health you can imagine” [25,26].Mini Mental State Examination (MMSE): the scale includes thirty questions that test memory, attention, calculation, naming, repeating, copying, understanding, and orientation in time and place. The cut-off level of the MMSE at ≥24 points represents improbable cognitive impairment versus ˂23 points indicating probable cognitive impairment. Evidence supporting the validity of this scale has been published previously [27]. The Polish standardization the MMSE scale is characterized by a high accuracy and reliability (Cronbach’s alpha coefficient of 0.88 for a clinical trial and 0.82 for healthy people) [28].Narrative interview and thematic analysis are specific methods aimed at obtaining information from the interlocutor about his/her subjective experiences [29,30]. In the study conducted here, this experience was connected with the experience of TAVI. The interview consisted of four parts. The first part involved creating a psychological relationship without discussing content related to the main topic. The second step was to stimulate the narration by asking a general question about the experiences and feelings of the person. The third phase was a proper narration on the main subject of the interview concerning the TAVI experience. Patients were encouraged to narrate their experience by the phrase: “Tell me about your TAVI experience, please.” Examples of questions that supplemented the patient’s story were: “How did you experience TAVI?” and “What thoughts and emotions have emerged during and after TAVI?” Additionally, after the end of the patient’s story, a fourth step followed, in which the researcher could clarify ambiguity and ask about narrative elements that raised doubts. At the end of the interview, the conversation returned to everyday matters. The qualitative method was used to describe the results in accordance with the thematic analysis of the emotions and beliefs related to TAVI and a sense of acceptance of TAVI.

### 2.3. Statistical Analysis

In the analysis, elements of descriptive statistics were applied. The distribution of the data was tested for normality using the Shapiro–Wilk test. The results are presented as the number of patients (percentage) or mean values with a standard deviation. We used an unpaired t-test or Mann–Whitney *U* test for the continuous variables and a chi-square test for the categorical variables, as appropriate. The strength of the associations between the health-related quality of life and depressiveness were measured using correlation analysis, and *p*-values below 0.05 were considered statistically significant. The statistical analysis was performed using Statistica 13 PL (StatSoft, Inc. Tulsa, OK, USA).

Data from the psychological interview were analyzed on the basis of the narrative interview and thematic analysis [29,30]. The patients’ statements were randomly selected and analysed by three competent judges who were the specialists in clinical psychology.

## 3. Results

### 3.1. Depressive Symptoms before and after TAVI

The mean level of depressive symptoms before TAVI (GDS-SF total) was 4.18 ± 2.84, and after TAVI it was 3.12 ± 2.52, and this difference achieved statistical significance. In accordance with the GDS-SF standards, this indicates that the levels of depressiveness of the patients were low, not reaching a clinically significant severity during the entire treatment process and follow-up period. Moreover, this depressiveness decreased after TAVI treatment. In terms of the individual symptoms of depression, a significant increase in the level of activity and number of interests was identified (GDS-SF 2: 0.47 ± 0.5 vs. 0.26 ± 0.44). On the other hand, a decrease in life satisfaction was described by the patients (GDS-SF 1: 0.25 ± 0.44 vs. 0.09 ± 0.29). The Mann–Whitney U-test was applied when the data were non-normally distributed (Table 2).

The further analysis showed that, in general, the frequency of depressive symptoms (with the GDS-SF ≥ 6 cut-off level for depression) before and after TAVI decreased from 20% (26 patients) to 3% (1 patient). In detail, in 50% (21 patients) of patients who were treated with TAVI, the severity of depressive symptoms diminished, in 29% (12 patients) the symptoms were stable, and in 21% (9 patients) they increased. Thus, if depressive symptoms were present before TAVI, their severity did not increase because of the treatment in 79% (33 patients) of the patients. Furthermore, patients without depressive symptoms before the treatment did not suffer from depressive moods due to TAVI.

### 3.2. Health-Related Quality of Life and Depressiveness

Before and after TAVI, all the correlations between the EQ-5D-3L descriptive items (from EQ-5D 1 to EQ-5D 5) and GDS-SF scores were positive, while those between the EQ-5D visual analogue scale (EQ VAS) and GDS-SF were negative (Table 3). This means that the better the quality of life described by the elderly patients was, the fewer depression symptoms they presented. The results oscillated from 0.62 to −0.18 (*p* < 0.05). The better the general health-related quality of life (EQ-total) described before TAVI was, the less depressed (GDS-SF total: r = 0.26), less helpless (GDS-SF 8: r = 0.34), less bored (GDS-SF 4: r = 0.21), less worthless (GDS-SF 12: r = 0.18), and happier (GDS-SF 7: r = 0.2) the patients were. They were also less worried about something bad happening to them when their overall quality of life was higher (GDS-SF 6: r = 0.23). In detail, the better the mobility of the patients before TAVI (EQ1 MO), the stronger their excitement to be alive (GDS-SF 11: r = 0.12), and the lower their sense of worthlessness were (GDS-SF 12: r = 0.27), the higher their level of energy (GDS-SF 13: r = 0.21), the lower their sense of hopelessness (GDS-SF 14: r = 0.19), and the lower their belief in being worse than others (GDS-SF 15: r = 0.21) were according to their descriptions. The better their ability to self-care was (EQ2 SC), the less boredom (GDS-SF 4: r = 0.25), better spirits (GDS-SF 5: r = 0.37), and lower depressiveness they perceived in general (GDS-SF total: r = 0.18). In relation to their usual activities (EQ3 UA), the more the patients could do, the lower their sense of hopelessness was (GDS-SF 14: r = 0.18). No significant correlations between perceived pain/discomfort (EQ4 PD) and depressiveness were found before TAVI. Meanwhile, the lower the level of anxiety/depression in patients was (EQ5 AD), the more they were characterized by less fear (GDS-SF 6: r = 0.19), a greater sense of happiness (GDS-SF 7: r = −0.19), and less feelings of helplessness (GDS-SF 8: r = 0.29). 

After TAVI, the correlations between health-related quality of life and depressive symptoms were stronger. After the treatment, the higher general quality of life (EQ-total) was correlated with lower levels of depressiveness (GDS-SF total: r = 0.48), hopelessness (GDS-SF 14: r = 0.48), memory problems (GDS-SF 10: r = 0.4), and helplessness (GDS-SF 8: r = 0.56), and a higher level of high spirits (GDS-SF 5: r = 0.39). In detail, the better the mobility after TAVI was (EQ1 MO), the less feelings of helplessness (GDS-SF 8: r = 0.36) and sense of hopelessness the patients experienced (GDS-SF14: r = 0.36). The better the ability to self-care after TAVI was (EQ2 SC), the lower the sense of hopelessness was (GDS-SF 14: r = 0.37). In the area of usual activities (EQ3 UA), the more the patients could do, the more activities and interests they had (GDS-SF 2: r = 0.4), and the less feelings of helplessness (GDS-SF 8: r = 0.36), the less memory problems (GDS-SF 10: r = 0.47), and less sense of hopelessness (GDS-SF 14: r = 0.58) they described. The less pain/discomfort (EQ4 PD) after TAVI the patients experienced, the greater high spirits (GDS-SF 5: r = 0.41), less feelings of helplessness (GDS-SF 8: r = 0.62), and less depressiveness they experienced in general (GDS-S total: r = 0.45). The lower the level of anxiety/depression was (EQ5 AD), the greater high spirits (GDS-SF 5: r = 0.46) and less fear (GDS-SF 6: r = 0.35) and general depressiveness (GDS-SF total: r = 0.43) the TAVI patients expressed. Negative correlations between the EQ VAS and GDS-SF before TAVI ranged between r = −0.24 for the level of high spirits (GDS-SF 5) and r = −0.18 for boredom (GDS-SF 4), with weak correlations between the EQ VAS and satisfaction with life (GDS-SF 1: r = −0.19) and general depressiveness (GDS-SF total: r = −0.19). Meanwhile, after TAVI, the correlations between the EQ VAS and HRQoL were stronger and ranged between r = −0.47 for the level of energy (GDS-SF 13) and sense of hopelessness (GDS-SF 14) and r = −0.4 for the feelings of helplessness (GDS-SF 8). Moderate correlations were also found between the EQ VAS and sense of worthlessness (GDS-SF 12: r = −0.44) and depressiveness in general (GDS-SF total: r = −0.49) after TAVI treatment.

### 3.3. Psychological Narrative Interview on Emotions and Beliefs after TAVI

Table 4 presents the results of a narrative psychological interview, with the patients’ statements about their mood after TAVI, divided into two categories: statements indicating a positive mood and acceptance of the TAVI results and statements indicating depressive symptoms. This analysis was performed based on 20 randomly selected patients using the competent judges method.

According to the thematic analysis, the themes most frequently mentioned in the patients’ statements were emotions and beliefs related to TAVI, the influence of TAVI on their mood and everyday activities and skills, cognitive functioning after TAVI, relationships with medical staff during the TAVI treatment process, the role of TAVI in family life, future joy in life and concerns, satisfaction with treatment, the acceptance of TAVI, and the sense of security (Table 4).

## 4. Discussion

This study measured the pre- and post-TAVI levels of depression and relationship between health-related quality of life (HRQoL) and depressiveness among the elderly who qualified for TAVI. In the majority (79%) of the patients, the level of depressiveness slightly decreased or was stable after the treatment. The better the HRQoL that the elderly person described was, the less depressive symptoms they presented. A qualitative analysis of the mood and beliefs of the TAVI population revealed that the patients presented both positive and depressive beliefs about their TAVI results.

The finding of the research demonstrating that approximately one in five patients who qualify for TAVI is affected by depressive symptoms is consistent with other studies’ results, showing a 20–30% rate of the diagnosis of depression in TAVI patients [5,19]. In this study, the TAVI procedure improved the mental state of the majority of the depressive patients after one year. This result is similar the findings of to Sun et al. [31], who found that the depression level was significantly improved 8 months after treatment, and Bäz et al. [19], whose results showed a persisting reduction in depression after 12 months. On the other hand, Lange et al. [32] noticed only moderate changes in the anxiety/depression of TAVI patients. To sum up, the present findings show that both the severity and the frequency of depressive symptoms after TAVI tend to decrease. 

After TAVI, a higher general quality of life was correlated with a lower level of depressiveness. The TAVI treatment can have a positive impact on the quality of life by improving the mood. The relationship between depressive symptoms and health-related quality of life can be understood bilaterally. On the one hand, HRQoL contributes to the severity of depressive symptoms. On the other hand, depressiveness has an impact on the HRQoL level. Boureau et al. [33] found that patients with depressive symptoms before the TAVI intervention showed mental QoL improvement after six months. In a group of patients with coronary artery disease, the previous analyses showed that despite successful surgical treatment [34,35], depressive symptoms significantly impaired the HRQoL. As depression is considered to be one of the risk factors for adverse events and mortality after cardiac surgery and changes in quality of life, clinicians should be vigilant with respect to the evaluation of the intensity of depressive symptoms before and after aortic stenosis treatment.

Patients reported higher activity levels and an increased number of interests after TAVI, which may indicate that they revealed themselves to other people and became more interested in the world around them. Lower levels of depressiveness and increases in activity and in the number of interests are consistent with clinical psychological knowledge. However, there is a high probability that the post-TAVI patients evaluated the factors regarded as the most valuable in their lives and personally important to them, other than their activities and interests. Surprisingly, however, the patients’ overall life satisfaction decreased. Satisfaction with life is a general form of assessment and is not oriented towards a particular domain [36], e.g., somatic symptoms. Taking into account the results obtained from the narrative interview and thematic analysis of the study, family life, satisfaction with treatment, and the sense of security were considered by the post-TAVI patients to be important and relevant in their subjective experiences. Advanced age, a multiplicity of past diseases, and the passage of one year after TAVI treatment may also have significant impacts on the overall life satisfaction of older people.

This study was the first, to our knowledge, to analyze the beliefs of older patients undergoing TAVI using the method of a narrative interview. The need for such a qualitative evaluation is in line with the previous research’s conclusions on a group of TAVI patients [5,12]. The study revealed that the most important areas of improvement in older adults who underwent TAVI and did not suffer from depressive symptoms were the possibility of performing self-care activities as part of daily living, better mobility, cognitive functioning amelioration, an elevated sense of security, and improved relationships with family members. On the other hand, in a group of patients with depressive symptoms after TAVI, the presence of aspects such as thoughts about the end of life, feelings of mental deterioration, fear of delirium syndrome, a lack of drive for life, and loneliness were observed. What is worth emphasizing is that some of the TAVI patients hesitated with respect to whether they considered this treatment suitable for them. Adverse events related to TAVI, the perception of being too old for the treatment, and the fear of being a burden on doctors and family are common concerns among patients. However, consultations (interviews, conversations) with healthcare professionals proved to be very effective in terms of handling patients’ uncertainties, highlighting the importance of the holistic approach in peri-procedural management and the significance of teamwork among health professionals. In a situation where depressive or cognitive decline symptoms are identified through a narrative interview or screening tool scores (GDS-SF and/or MMSE), more detailed neuropsychological and geriatric assessments should be taken into consideration. Furthermore, preexisting co-morbidities are predictive of poor outcomes in the TAVI population [12], and under-recognized cognitive deficits (e.g., mild cognitive impairment) may increase the risk of post-TAVI adverse results [37]. Thus, emotional and mental state assessment prior to TAVI may be helpful for the identification of older patients who can benefit most from TAVI and those who need psychological help and/or anti-depressive or neurological treatment before making the final decision between TAVI or a conservative treatment. 

## 5. Conclusions

In summary, the results obtained and the literature reviewed indicate the need for interdisciplinary and comprehensive hospital care before and after TAVI. Health-related quality of life should be considered as an important aspect of depressive symptoms in older adults undergoing TAVI. Special care should be provided to TAVI patients who exhibit depressive symptoms in order to increase their chances of improving the HRQoL throughout the TAVI treatment and its outcomes. These results underscore the importance of routine screening for depression integrated into the preoperative period for all older (>65 years old) TAVI patients and the reassessment of changes in depressiveness during follow-up.

### Study Limitations

The first and the main limitation of this study is the small number of patients in the post-TAVI group. There was also a potential bias caused by the patients lost during follow-up. Secondly, even though the GDS-SF is a validated and widely used instrument to screen for depression, there was no confirmatory testing by a psychiatrist, which could have acted as a source of the misdiagnosis of a depressive state.

## Figures and Tables

**Figure 1 healthcare-10-02211-f001:**
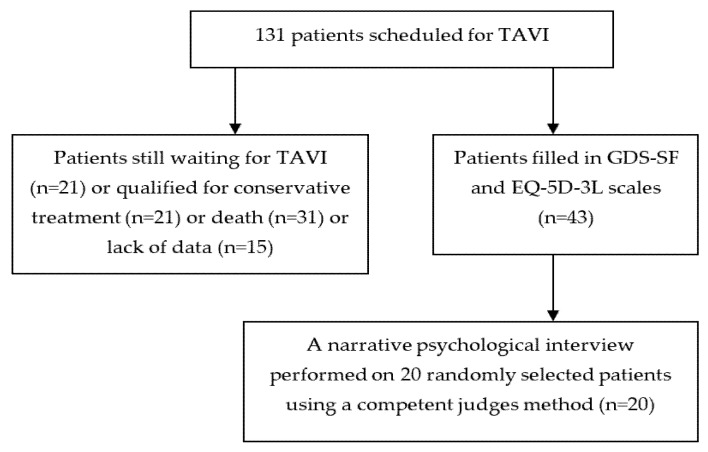
Flow chart of patients with severe aortic stenosis qualified for TAVI.

**Table 1 healthcare-10-02211-t001:** Demographic and clinical characteristics of the elderly patients scheduled for TAVI.

	Total (*n* = 131)
Age	82.1 (6.1)
Women	83 (63%)
Education level	9.5 (9.8)
Hypertension	39 (30%)
Coronary artery disease	34 (26%)
Diabetes	16 (12%)
Stroke	5 (4%)

**Table 2 healthcare-10-02211-t002:** The results regarding emotional functioning before and after TAVI (Mann–Whitney *U*-test, * *p* < 0.05).

GDS-SF Item	Sum of Rangs before TAVI *n* = 131	Sum of Rangs after TAVI *n* = 43	U	Z	*p*	Z Corrected	*p*
GDS-SF 1	11,910	3315	2369	1.56	0.12	2.20	0.03 *
GDS-SF 2	12,053.5	3171,5	2225.5	2.06	0.04	2.42	0.02 *
GDS-SF 3	11,587.5	3637.5	2691.5	0.43	0.66	0.75	0.45
GDS-SF 4	11,475	3750	2804	0.04	0.97	0.06	0.95
GDS-SF 5	11,610	3615	2669	0.51	0.61	0.77	0.44
GDS-SF 6	11,854	3371	2425	1.36	0.17	1.58	0.12
GDS-SF 7	11,738	3487	2541	0.96	0.34	1.49	0.14
GDS-SF 8	11,582.5	3642.5	2696.5	0.42	0.68	0.56	0.57
GDS-SF 9	11,532.5	3692.5	2746.5	0.24	0.81	0.29	0.77
GDS-SF 10	11,252	3973	2606	−0.73	0.46	−0.89	0.37
GDS-SF 11	11,568.5	3656.5	2710.5	0.37	0.71	0.72	0.47
GDS-SF 12	11,633	3592	2646	0,59	0.55	1.07	0.28
GDS-SF 13	11,627.5	3597.5	2651.5	0.57	0.57	0.69	0.49
GDS-SF 14	11,610	3615	2669	0.51	0.61	0.77	0.44
GDS-SF 15	11,627.5	3597.5	2651.5	0.57	0.57	0.69	0.49
GDS-SF total	11,921	2957	2054	2.41	0.02	2.43	0.02 *

Abbreviations: GDS-SF 1—satisfaction with life; GDS-SF 2—activities and interests; GDS-SF 3—emptiness of life; GDS-SF 4—boredom; GDS-SF 5—good spirits; GDS-SF 6—fear; GDS-SF 7—sense of happiness; GDS-SF 8—feeling of helplessness; GDS-SF 9—preference for staying at home instead of going out; GDS-SF 10—memory problems; GDS-SF 11—excitement to be alive; GDS-SF 12—sense of worthlessness; GDS-SF 13—level of energy; GDS-SF 14—sense of hopelessness; GDS-SF 15—belief in being worse than others; GDS-SF total—general level of depressiveness.

**Table 3 healthcare-10-02211-t003:** Correlations between the GDS-SF and EQ-5D-3L items before and after TAVI (* *p* < 0.05).

	before TAVI (*n* = 122)						
GDS-SF Item	EQ-1 MO	EQ-2 SC	EQ-3 UA	EQ-4 PD	EQ-5 AD	EQ-Total	EQ VAS
GDS-SF 1	−0.08	−0.005	0.06	0.009	−0.03	0.16	−0.19 *
GDS-SF 2	−0.01	−0.04	−0.12	0.04	0.03	0.1	−0.03
GDS-SF 3	−0.05	0.1	−0.09	−0.04	0.11	0.08	−0.08
GDS-SF 4	0.04	0.25 *	0.16	0.15	0.16	0.21 *	−0.18 *
GDS-SF 5	−0.001	0.37 *	−0.08	0.05	−0.08	0.15	−0.24 *
GDS-SF 6	0.11	0.1	0.11	0.04	0.19 *	0.23 *	−0.03
GDS-SF 7	0.01	0.04	0.05	0.03	0.19 *	0.20 *	−0.13
GDS-SF 8	0.07	0.03	0.14	0.08	0.29 *	0.34 *	−0.135
GDS-SF 9	−0.04	0.06	0.01	0.1	0.01	0.14	0.08
GDS-SF 10	−0.02	−0.02	0.03	−0.06	−0.07	0.15	0.02
GDS-SF 11	0.20 *	−0.04	0.07	0.06	0.04	0.12	−0.02
GDS-SF 12	0.27 *	−0.02	0.12	0.06	0.09	0.18 *	0
GDS-SF 13	0.21 *	−0.05	0.03	0.03	0.06	0.11	−0.007
GDS-SF 14	0.19 *	0.02	0.18 *	0.05	0.12	0.17	−0.01
GDS-SF 15	0.21 *	−0.01	0.1	0.06	0.06	0.13	−0.003
GDS-SF total	−0.03	0.18 *	0.08	0.07	0.17	0.26 *	−0.19 *
	**after TAVI (*n* = 37)**						
GDS-SF Item	EQ-1 MO	EQ-2 SC	EQ-3 UA	EQ-4 PD	EQ-5 AD	EQ-Total	EQ VAS
GDS-SF 1	0.11	0.17	0.07	0.18	0.31	0.26	−0.06
GDS-SF 2	−0.05	0.14	0.41 *	0.08	0.24	0.25	−0.32
GDS-SF 3	0.01	0.12	0.21	0.09	0.2	0.19	−0.09
GDS-SF 4	0.01	0.04	0.11	0.26	0.32	0.25	0.02
GDS-SF 5	0.16	0.04	0.11	0.41 *	0.46 *	0.39 *	−0.16
GDS-SF 6	−0.26	−0.16	−0.22	0.13	0.35 *	−0.02	0.07
GDS-SF 7	0.11	−0.12	−0.14	0.18	−0.05	0.004	−0.16
GDS-SF 8	0.36 *	0.21	0.36 *	0.62 *	0.25	0.56 *	−0.40 *
GDS-SF 9	0.02	−0.15	−0.12	0.2	0.1	0.03	−0.19
GDS-SF 10	0.24	0.18	0,47 *	0.12	0.2	0.40 *	−0.28
GDS-SF 11	0.25	0.26	0.14	0.06	0.25	0.29	−0.27
GDS-SF 12	−0.09	0.17	0.28	0.18	−0.05	0.13	−0.44 *
GDS-SF 13	0.19	0.04	0.14	0.24	0.14	0,24	−0.47 *
GDS-SF 14	0.36 *	0.37 *	0.58 *	0.27	0.04	0.48 *	−0.47 *
GDS-SF 15	0.08	−0.26	−0.17	−0.02	0.13	−0.04	−0.05
GDS-SF total	0.22	0.11	0.29	0.45 *	0.43 *	0.48 *	−0.45 *

Abbreviations: EQ-5D-3L comprises five dimensions: mobility (MO), self-care (SC), usual activities (UA), pain/discomfort (PD), and anxiety/depression (AD). EQ VAS—visual analogue scale; GDS-SF total—general level of depressiveness; GDS-SF 1—satisfaction with life; GDS-SF 2—activities and interests; GDS-SF 3—emptiness of life; GDS-SF 4—boredom; GDS-SF 5—good spirits; GDS-SF 6—fear; GDS-SF 7—sense of happiness; GDS-SF 8—feeling of helplessness; GDS-SF 9—preference for staying at home instead of going out; GDS-SF 10—memory problems; GDS-SF 11—excitement to be alive; GDS-SF 12—sense of worthlessness; GDS-SF 13—level of energy; GDS-SF 14—sense of hopelessness; GDS-SF 15—belief in being worse than others.

**Table 4 healthcare-10-02211-t004:** Examples of patients’ statements about their mood after TAVI based on a psychological narrative interview and thematic analysis.

Statements Indicating a Positive Mood and Acceptance of TAVI Results	Statements Indicating DepressiveSymptoms
“I feel normal. I am not nervous. If not for this procedure, I would be gone. I hit good hands” (69 years old)	“I had a crisis after TAVI—I was cold and weakened and I cried, which is unusual for me” (85 years old)
“I appreciate the procedure so much! I was given a second life” (75 years old)	“Due to hospitalization, my psyche deteriorated” (86 years old)
“The mood is good. I go out to my great-granddaughters, and there are 39 steps, and I walk up the stairs without any problems. I am so savvy that I do not want to spoil this surgery. So, I have myself these six months after TAVI, and I would like to hang curtains at home!” (64 years old)	“My mood—I cannot stop thinking about the end of my life after the surgery. I am a neurotic person. My husband was ill and it also contributed.” (85 years old)
“It’s much better—as different as day and night. I am not even nervous, I sleep well while before the surgery I could not sleep, I had a low mood, nerves and crying” (78 years old)	“My hospital stay was associated with some bad emotions. I had these symptoms [delirium syndrome]. I could not sleep for many nights” (91 years old)
“I have a good humor. If it was not for the surgery, I would be in my grave. I thought ‘what do I need it [TAVI] for?!’ I had a grudge against my doctor, why she put me there [for surgery]. It was a good choice.” (77 years old)	“Mentally, I feel terrible because I fell during rehabilitation and I have an injury to my left knee. I am terrified of the prospect” (77 years old)
“TAVI improved my heart a lot. I have no weaknesses, I do not lose consciousness, as I had before. Everything is fine now. I only suffer from the gait instability” (79 years old)	“When it comes to my heart, I feel great. I do not suffer from dyspnea and tiredness. Staying at home depresses me” (86 years old)
“I could neither put on the socks, nor wash myself [before TAVI] and now I can do everything: go out, chop wood, scythe mow next to the currents, feed the chickens”, “My daughter is dissatisfied that I want to do so much after treatment” (77 years old)	“I feel like a looser. I have no drive for life, no desire. I would prefer they locked me [in a retirement home] and only gave me water” (76 years old)
“My mood is balanced, I do not cry. I would like God to keep me alive until summer because I have two ceremonies—the baptism and communion of my grandsons” (86 years old)	“I am in a bad mood. I feel terribly sad. I do not know a cause of that. Before that, the mood was different. Now I cannot take care of my grandchildren” (85 years old)
“I have come to terms mentally with all of this. What God gives, man will not change. My life was extended. I am very grateful to the doctors. Otherwise, I would not be alive” (77 years old)	“I do not want to live. My nerves get worse every year. The neighbors visit me and it saves me” (87 years old)
“It is much more advantageous with emotions. Before the operation I could not walk and I was resigned” (85 years old)	
“My emotions are good. Recently, my wife and I were at the seaside. I am still working. I am a tailor. Recently I was sewing a suit” (86 years old)	

## Data Availability

The data that support the findings of this study are available from the corresponding author upon reasonable request.

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
