# Peer review of "Quality of Life and Depressive Symptoms in Transcatheter Aortic Valve Implementation Patients—A Cross-Sectional Study"

_healthcare, 2022, doi:10.3390/healthcare10112211_

Round 1

Reviewer 1 Report

Add research design to the title

Do not use abbreviations in the abstract.

Add research design to the methodology section. 

Do you have Cronbach Alpha for each instrument?

Author Response

Response to Reviewer 1 Comments

Dear Sir or Madam,

on behalf of the authors of the paper we want to express our gratitude for all comments and suggestions.

Point 1: Add research design to the title;

Response 1:  We added a “research design” to the title.

Point 2: Do not use abbreviations in the abstract.

Response 2:  We removed all the abbreviations from the abstract.

Point 3: Add research design to the methodology section. 

Response 3:  We added a “research design” to the methodology section.

Point 4: Do you have Cronbach Alpha for each instrument?

Response 4:  In the Polish standardization the Cronbach’s Alpha coefficient for GDS-SF is 0.92, for MMSE 0.88 for a clinical trial and 0.82 for healthy people. Unfortunately, I could not find this data for EQ-5D-3L. we wrote an e-mail to the author of the Polish adaptation of EQ-5D-3L asking for this information, but so far we have not received an answer.

We hope that introduced changes meet your requirements.

Kind regards,

authors of the manuscript

Reviewer 2 Report

The study performed by authors “Depressive symptoms before and after ranscatheter aortic valve implementation in older adults and relation between quality of life and depressiveness in TAVI patients.”, there are some minor points to be further addressed.

#  Minor comments

-          The language requires professional editing in many places.

-          Title needs to revise; it is long and boring.

-          Abstract need to revise according to journal guideline.

# Result:

-       Results: Prepare tables smoother and more friendly for readers. They need to compare in some ways, so it should be suggested to use special symbols to show comparisons and define them.

#  Discussion:

Discussion is poor and needs to be better presented.

# References:

The references should be in accordance with the journal's guideline and in the same format.

Author Response

Response to Reviewer 2 Comments

Dear Sir or Madam,

on behalf of the authors of the paper we want to express our gratitude for all comments and suggestions.

Point 1: The language requires professional editing in many places.

Response 1:  According to your recommendations the language was edited.

Point 2: Title needs to revise; it is long and boring.

Response 2:  We changed  the title of the manuscript making it shorter and I hope more interesting. A new title is “Quality of life and depressive symptoms in transcatheter aortic valve implementation patients- a cross-sectional study”

Point 3: Abstract need to revise according to journal guideline.

Response 3:  We changed the abstract according to journal quidelines. A new abstract: Quality of life is an important factor influencing mood. In a group of elderly undergoing valve implementation or surgical aortic valve replacement, one in 3 has depressive symptoms. The aims of the study: 1. to evaluate the impact of health related-quality of life on depressive symptoms in the elderly patients undergoing TAVI, 2. to analyse beliefs according to TAVI. Methods: 131 elderly (mean age: 82.1 ± 6.1 years) scheduled for TAVI completed Geriatric Depression Scale, EQ-5D-3L and Mini Mental State Examination. 43 patients completed the questionnaires after the treatment. The narrative interview analyses were performed in 20 randomly selected patients after TAVI. Results: The mean level of general depressive state before TAVI was 4.19 ± 2.83, and after 3.12 ± 2.52 (p=0.02) and the frequency decreased from 20% to 3%. An increase in the level of activity and number of interests and a decrease in life satisfaction were found. The higher general quality of life, the lower level of depressiveness before and after TAVI (r=0.26 vs. r=0.48; p<0.05). Conclusions: Patients before and after TAVI differed in depressive symptoms, their intensity and frequency. These results underscore the importance of screening for depression at baseline and reassessing changes in depressiveness during follow-up.

Point 4: Results: Prepare tables smoother and more friendly for readers. They need to compare in some ways, so it should be suggested to use special symbols to show comparisons and define them.

Response 4:  In the tables we used the “*” abbreviation for  p<0.05 and letter abbreviations for the scale items. I would be grateful for a hint or example on how to improve the tables.

Point 5: Discussion is poor and needs to be better presented.

Response 5:  According to your recommendations we expanded the discussion.

Point 6: The references should be in accordance with the journal's guideline and in the same format.

Response 6:  we changed the references according to journal quidelines.

We hope that introduced changes meet your requirements.

Kind regards,

authors of the manuscript

Reviewer 3 Report

Manuscript ID: healthcare-1972145

The article provides useful information on " Depressive symptoms before and after transcatheter aortic valve implementation in older adults and the relation between quality of life and depressiveness in TAVI patients". The manuscript needs some revisions, because there are some aspects of the work that should be corrected and improved. Please, review the following recommendations:

Line 3: in title: Add "the" before " relation "

Lines 15,36, 40, 59, 192, 215, 282, 325, 328: Change " health related" to " health-related"

Line 29: Change " prior " to " prior to"

Lines 35, 322: Change " accompanied with" to " accompanied by"

Line 87: Add "The" before "Level "

Line 87: Change "in male than in female" to "in males than in females"

Line 100: Delete space before " patients"

Line 103: Delete space after "treatment"

Line 136: Delete "which"

Line 136: Change "a specific method" to "specific methods"

Line 149: Delete space after "matters"

Line 170: Add "," after "Moreover"

Line 196: Delete "was"

The discussion section should be improved.

Lines 264-265, 270: Change "qualified to" to "qualified for"

Line 265: Change "In majority" to "In the majority"

Line 274: Change "at al." to "et al."

Lines 280-284: Rewrite these sentences

Lines 305-308: Rewrite these sentences

Line 325: Change " related with" to " related to"

It is preferable to shorten the "5. Conclusions"

Insert the correct format style for journals in the references in the text and references list.

Author Response

Response to Reviewer 3 Comments

Dear Sir or Madam,

on behalf of the authors of the paper we want to express our gratitude for all comments and suggestions.

Point 1: Line 3: in title: Add "the" before " relation "

Lines 15,36, 40, 59, 192, 215, 282, 325, 328: Change " health related" to " health-related"

Line 29: Change " prior " to " prior to"

Lines 35, 322: Change " accompanied with" to " accompanied by"

Line 87: Add "The" before "Level "

Line 87: Change "in male than in female" to "in males than in females"

Line 100: Delete space before " patients"

Line 103: Delete space after "treatment"

Line 136: Delete "which"

Line 136: Change "a specific method" to "specific methods"

Line 149: Delete space after "matters"

Line 170: Add "," after "Moreover"

Line 196: Delete "was"

Lines 264-265, 270: Change "qualified to" to "qualified for"

Line 265: Change "In majority" to "In the majority"

Line 274: Change "at al." to "et al."

Lines 280-284: Rewrite these sentences

Lines 305-308: Rewrite these sentences

Line 325: Change " related with" to " related to"

Response 1:  We rewrote the sentences and made the suggested changes to all of the lines mentioned.

Point 2: The discussion section should be improved.

Response 2:  According to your recommendations we expanded the discussion.

Point 3: It is preferable to shorten the "5. Conclusions"

Response 3:  we shortened the “5. Conclusions” part of the manuscript.

Point 4: Insert the correct format style for journals in the references in the text and references list.

Response 4: We changed the format style for journals in the references in the text and references list according to journal quidelines.

We hope that introduced changes meet your requirements.

Kind regards,

authors of the manuscript

Reviewer 4 Report

In the present manuscript, Katarzyna Olszewska-Turek and Barbara Betkowska-Korpala, focus on depressive symptoms before and after transcatheter aortic valve replacement. With the present work they illuminate an interesting and yet less discussed topic.  As maybe  expected depressive symptoms chenged in the presented patient cohort after TAVI. Patients described a higher level of activity and number of interests. However, surprisingly the overall life satisfaction decreased.

1.) How the authors explain this finding? One would expect that with better activity levels and increased interest, life satisfaction would also increase.

2.)Did you had a closer look to the perioperative course of the patients reporting statements indicating deepressive symptoms. Did intraoperative and postoperative complications had an impact on patients subsequent satisfaction, activity, and quality of life?

3.) The authors mentioned that 131 patients were scheduled for TAVI. In the end only 43 patients were included in the questionaire arm and 20 in the interview arm. What happend to the rest? Maybe you can make a small diagram showing the initially schedulded and finally included patients and point out reasons for exclusion.

4.) Do the authors think a routine screening for depressive symptoms should be integrated in the preoperative workup of all older (>65years) TAVI patients?

Author Response

Response to Reviewer 4 Comments

Dear Sir or Madam,

on behalf of the authors of the paper we want to express our gratitude for all comments and suggestions.

Point 1: How the authors explain this finding? One would expect that with better activity levels and increased interest, life satisfaction would also increase.

Response 1:  In a discussion we added a paragraph with a probable explanation of the results: “Patients reported higher activity levels and an increased number of interests after TAVI, which may indicate that they revealed themselves to other people, and became more interested in the world around them. Lower level of depressiveness, and an increase in activity and in number of interests are consistent with clinical psychological knowledge. However, there is a high probability that post-TAVI patients evaluated factors regarded as most valuable in their lives and personally important to them, other than activities and interests. Surprisingly, however, patients' overall life satisfaction decreased. Satisfaction with life is a general assessment, and is not oriented to a particular domain [36], e.g. somatic symptoms. Taking into account the results obtained from the narrative interview and thematic analysis of the study: family life, satisfaction with treatment, and sense of security were considered by post-TAVI patients to be important and relevant in their subjective experiences. Advanced age, multiplicity of past diseases, and the passage of one year after TAVI treatment may also have a significant impact on the overall life satisfaction of older people.

Point 2: Did you had a closer look to the perioperative course of the patients reporting statements indicating deepressive symptoms. Did intraoperative and postoperative complications had an impact on patients subsequent satisfaction, activity, and quality of life?

Response 2:  We did not have an opportunity to have a closer look to the perioperative course of the patients reporting statements indicating depressive symptoms because the TAVI treatment of all patients was not performed in our hospital, but in another hospital. Your question is of great interest to us and worthy of further investigation in another study, taking into account the results of other studies on postoperative complications including the impact of postoperative delirium on the quality of life and/ or mortality of patients.

Point 3: The authors mentioned that 131 patients were scheduled for TAVI. In the end only 43 patients were included in the questionaire arm and 20 in the interview arm. What happend to the rest? Maybe you can make a small diagram showing the initially schedulded and finally included patients and point out reasons for exclusion.

Response 3:  We added Figure 1. Flow chart of patients with severe aortic stenosis qualified for TAVI.

Point 4: Do the authors think a routine screening for depressive symptoms should be integrated in the preoperative workup of all older (>65years) TAVI patients?

Response 4:  Yes, we want to promote a routine screening for depressive symptoms in the preoperative time of all older (>65 years old) TAVI patients and reassessment of changes in depression during follow-up.

We hope that introduced changes meet your requirements.

Kind regards,

authors of the manuscript